# Simulating the Permeation of Toxic Chemicals through Barrier Materials

**DOI:** 10.3390/membranes14090183

**Published:** 2024-08-24

**Authors:** Alex Bicket, Vivian Lau, Jules Thibault

**Affiliations:** 1Department of Chemical and Biological Engineering, University of Ottawa, Ottawa, ON K1N 6N5, Canada; abick044@uottawa.ca; 2Defence Research and Development Canada, Ralston, AB T1A 8K6, Canada; vivian.lau@drdc-rddc.gc.ca

**Keywords:** personal protective equipment, Fickian diffusion, numerical simulation, liquid coverage pattern, material properties

## Abstract

Chemical warfare agents that are liquids with low vapor pressure pose a contact hazard to anyone who encounters them. Personal protective equipment (PPE) is utilized to ensure safe interaction with these agents. A commonly used method to characterize the permeability of PPE towards chemical weapons is to apply droplets of the liquid agent to the surface of the material and measure for chemical breakthrough. However, this method could produce errors in the estimated values of the transport properties. In this paper, we solved numerically the three-dimensional cylindrical Fick’s second law of diffusion for a liquid permeating through a non-porous rubbery membrane to determine the time the permeating species will emerge on the other side of the polymer membrane. Simulations of different amounts of surface area coverage and the geometries of permeate on the membrane surface indicated that incomplete surface area coverage affects the estimation of the transport properties, making the experimentally determined transport properties unsuitable for predictive use. We simulated different permeation values to determine the factors that most influenced the estimation error and if the error was consistent over different permeate–membrane combinations. Finally, a method to correct the experimentally determined permeability is suggested.

## 1. Introduction

When a hazardous chemical is released, it often requires people to access the area and put themselves at risk of exposure to the hazard to rescue injured people, investigate the event and its impact, or conduct a clean-up of the hazardous contamination. Elastomeric membranes are used as barrier materials in personal protective equipment (PPE) to protect against skin contact with toxic chemicals, including chemical warfare agents [1]. Non-porous rubbery polymers such as nitrile rubber, silicone rubber, and butyl rubber are common PPE materials, especially for gloves [2] where their flexibility gives the wearer dexterity while still being afforded a good degree of protection. In addition to functional properties, such as flexibility and weight, barrier materials are selected based on their chemical resistance, which is evaluated by measuring breakthrough, permeation, and degradation of the material when exposed to chemicals of interest [1]. Breakthrough time is one measure of how well a given barrier will protect against a chemical hazard, measuring the time it takes a chemical to completely permeate through the material [3], and is used to rate how long a piece of protective equipment will protect the wearer against that chemical.

However, transport properties are difficult to predict and are reliant on the individual system (permeate, membrane, and temperature), requiring physical experiments to derive the protective properties of the materials. In the case of diffusion through non-porous polymer membranes, the solution diffusion model is used to describe the motion of molecules of the permeate into and through the membrane [4,5,6]. Experimental procedures such as those described by ASTM F739, ASTM D6978, EN 374, EN 16523, ISO 6529, or US Army TOP 08-2-501A [7,8] are used to characterize the permeation of liquids through protective materials and determine transport properties in combination with the time-lag method of determining diffusivity using the steady-state flux of permeate through a membrane, as described by Dalton, Frisch and others [6,8,9,10].

Test methods used to evaluate PPE materials for chemical warfare defense applications involve placing individual droplets of a chemical agent on a circular sample of the test fabric and sampling the clean side of the membrane to detect the breakthrough of the chemical [8,11]. These tests can be resource-intensive and difficult to perform with highly toxic chemical permeates. However, simulation of the diffusion of molecules through dense polymer membranes is computationally intensive and difficult to use to predict transport properties [6,12].

An alternative approach is to apply experimentally obtained transport properties to computational models of membrane permeation. However, the experimentally obtained transport properties measured in standardized tests may not be appropriate. Rivin et al. [11] suggested that test methods using droplets of chemical permeant produce errors in the estimation of the transport properties derived through such testing suspected to be caused by the system not reaching steady-state diffusion or leading to transport properties different from those that would be obtained in the case of flooded permeation cells. Indeed, in this work, it is shown that transport properties obtained from experiments with partial coverage of the test membrane may be inaccurate. As a result, transport properties obtained from such experiments need to be interpreted with caution when used for PPE selection to protect against hazardous chemicals.

The objective of this study is to perform a series of numerical experiments to assess the impact of a partially covered membrane surface on the estimation of the membrane transport properties determined using the time-lag method. The impacts of partial coverage were assessed in conjunction with changing membrane thickness, changing the diffusivity of the permeate in the membrane, and changing the coverage geometry of the permeate on the surface of the membrane. The motivation for using partial membrane coverage is to minimize the amount of toxic liquid used in a permeation experiment. It is anticipated that the series of numerical simulations will allow us to find a correction factor that could be used to recover the actual membrane transport properties from those obtained experimentally. With a good estimate of the transport properties, the numerical model could then be used to examine various permeation scenarios.

## 2. Materials and Methods

### 2.1. Experimental Permeation System

The numerical experiments performed in this investigation are designed to model a typical diffusion cell for testing the permeation of hazardous chemicals [7,8,11,13]. A circular membrane swatch of the test membrane material is used and permeate is applied to one side of the membrane. As the permeate diffuses through the membrane, the amount of permeate found on the receptor side is quantified as a function of time to determine the transport properties of the membrane. The numerical experiments conducted were used to simulate the permeation of liquids through non-porous polymer membranes. Figure 1 shows a representative diffusion cell which is used to determine experimentally the transport properties of the liquid moving across a membrane [7,8,11,13,14].

### 2.2. Numerical Simulations

In this investigation, the three-dimensional Fick’s second law of diffusion Equation (1) was solved by finite differences using a program, written in Fortran programming language, developed by the authors for this application to determine the rate of permeation through a cylindrical dense membrane (Figure 2a), as typically incorporated within a permeation cell.
(1)$$\frac{\partial c}{\partial t}=D\left[\frac{\partial^{2}c}{\partial z^{2}}+\frac{\partial^{2}c}{\partial r^{2}}+\frac{1}{r}\frac{\partial c}{\partial r}+\frac{1}{r^{2}}\frac{\partial^{2}c}{\partial \phi^{2}}\right]$$

Equation (1) may be reduced to a one-dimensional partial differential equation in the permeation direction z when the feed side of the membrane is completely flooded. When the feed side of the membrane is partially covered, the three-dimensional diffusion equation must usually be solved. In this investigation, the finite difference in the three-directional mass transfer equation will always be solved to determine the temporal variation in the concentration within the membrane.

Equation (1) was discretized into a sufficiently large number of mesh points to generate small three-dimensional elements as shown in Figure 2b to achieve mesh independence. The discretized equation for a central mesh point is given in Equation (2) in its explicit form.
(2)$$\frac{c_{i,j,k}^{t+\Delta t}\;-\;c_{i,j,k}^{t}}{\Delta t}=D\left[\begin{array}{l} \frac{c_{i-1,j,k}^{t}\;-\;2c_{i,j,k}^{t}+c_{i+1,j,k}^{t}}{\Delta z^{2}}+\\ \frac{c_{i,j-1,k}^{t}\;-\;2c_{i,j,k}^{t}+c_{i,j+1,k}^{t}}{\Delta r^{2}}+\frac{1}{(j-1)\Delta r}\frac{c_{i,j+1,k}^{t}\;-\;c_{i,j-1,k}^{t}}{2\Delta r}\\ +\frac{1}{{\left((j-1)\Delta r\right)}^{2}}\frac{c_{i,j,k-1}^{t}\;-\;2c_{i,j,k}^{t}+c_{i,j,k+1}^{t}}{\Delta \phi^{2}} \end{array}\right]$$

The indices (*i*,*j*,*k*) refer to (*z*,*r*,*ϕ*) directions divided into (*NI*,*NJ*,*NK*) meshes. This equation calculates the internal concentration *c_i,j,k_* at mesh point (*i*,*j*,*k*) at the next time step Δ*t*, i.e., at time *t* + Δ*t*, based on the concentration at mesh point (*i*,*j*,*k*) and its six neighboring mesh points (Figure 2b) at time *t*. Equation (2) is valid for all internal mesh points describing the membrane. It is assumed that the dense membrane is isotropic, implying the membrane has a uniform composition structure throughout with constant diffusivity *D* and solubility *S*. In addition, it is assumed that the diffusivity and solubility are not functions of concentration and the system is isothermal.

To completely define the problem, it is necessary to provide the initial condition and six boundary conditions, two for each dimension. It is assumed that initially, the concentration of the migrating species throughout the membrane is zero (Equation (3)).
(3)$$\mathrm{At}\;t=0,\;c_{i,j,k}^{t=0}=0\;\forall i,j,k$$

At time zero, the liquid permeating species is added to the top surface of the membrane where it progressively dissolves into the dense membrane and migrates to the permeate side. The boundary conditions for the *z*-direction are given in Equations (4) and (5). Equation (4) simply states that the species concentration (*c*) at the top surface within the membrane is in instantaneous equilibrium with the concentration (*C*) of the liquid coverage outside the membrane. *S* is the membrane solubility of the migrating species. Equation (5) assumes that the concentration of permeate at the surface of the membrane on the receptor side is zero or close to zero. While this will not be strictly the case, we can assume that in a permeation cell where a liquid receptor solution is used instead of a sweep gas, the solvent has a high enough affinity for the permeate that the concentration at the membrane surface is equivalent to the concentration of permeate in the bulk solvent. In a diffusion cell where the permeant is sampled on the receptor side as a vapor, a constant gas flow is used to carry the permeant vapor to the detector [11]. As the rates of diffusion in the modeled system are generally very low and the levels of permeate that indicate breakthrough are generally very small, then the assumption is still valid.
(4)$$\mathrm{At}\;z=0,\;c_{i=1,j,k}^{t}=SC_{i=1,j,k}^{t}\;\forall j,k$$
(5)$$\mathrm{At}\;z=L,\;c_{i=NI,j,k}^{t}=0\;\forall j,k$$

The boundary conditions for the radial direction are given in Equations (6) and (7). Equation (6) states that symmetry prevails at the center (*r* = 0) of the membrane. The second term in the *r*-direction of Equation (1) is undetermined at the center since both the partial derivative and the radius are both equal to zero. However, using L’Hôpital’s rule, this term is easily equated to the second partial derivative concerning *r* and the radial term can be easily discretized. It is important to note that for the symmetry condition at the center of the membrane to prevail, the liquid must be distributed symmetrically on the surface of the membrane, especially when a multiple-drop pattern is used. Equation (7) simply states that at the outside circumference of the membrane, an impermeability condition prevails.
(6)$$\mathrm{At}\;r=0,\;\frac{\partial c_{i,j=1,k}^{t}}{\partial r}=0\;\forall i,k$$
(7)$$\mathrm{At}\;r=R,\;\frac{\partial c_{i,NJ,k}^{t}}{\partial r}=0\;\forall i,k$$

The boundary conditions for the angular direction (*ϕ*) are given in Equations (8) and (9). Both boundary conditions assumed symmetry at *ϕ* equal to zero and some other angle where symmetry prevails. This angle could be 2*π* if symmetry does not prevail at a smaller angle. In this investigation, *π*/2 and 2*π* were used depending on the case study. One particular aspect of the angular term of Equation (1) is the singularity at the center point of the membrane. Few techniques have been proposed to deal with this singularity [15]. In this investigation, at the center point (*j* = 1), the zero radius was replaced with 0.1% of the first radial mesh point Δ*r*.
(8)$$\mathrm{At}\;\phi =0,\;\frac{\partial c_{i,j,k=1}^{t}}{\partial \phi }=0\;\forall i,j$$
(9)$$\mathrm{At}\;\phi =\phi_{1},\;\frac{\partial c_{i,j,k=NK}^{t}}{\partial \;\phi }=0\;\forall i,j$$

The average upstream and downstream fluxes were obtained from the concentration gradients at the upstream and downstream fluid–membrane interfaces using Equations (10) and (11). Convergence to the steady permeation rate was assumed to be reached when the average fluxes (*J*) at the upstream and downstream surfaces were the same, within 0.01%. The downstream average flux allows calculation of the rate of accumulation of the migrating species on the permeate side. Under steady state, the rate of accumulation becomes constant and the extrapolation to the time axis of the constant accumulation curve as a function of time provides the time lag (*θ*) from which the effective diffusivity can be determined, see Equation (12).
(10)$$J_{z=0}=\frac{1}{\pi R^{2}}\int_{0}^{R}\;\int_{0}^{2\pi }-D{\frac{\partial \;c}{\partial \;z}|}_{z=0}rdr\;d\phi \;(\mathrm{Upstream})$$
(11)$$J_{z=L}=\frac{1}{\pi R^{2}}\int_{0}^{R}\;\int_{0}^{2\pi }-D{\frac{\partial \;c}{\partial \;z}|}_{z=L}rdr\;d\phi \;(\mathrm{Downstream})$$
(12)$$\theta =\frac{L^{2}}{6D_{eff}}$$
where *R* is the radius of the membrane, *D_eff_* is the effective diffusivity of the migrating species in the membrane, and *L* is the membrane thickness. Note that an apparent or effective diffusivity (*D_eff_*) has been used, Equation (12), instead of the intrinsic diffusivity (*D*) because the measured diffusivity via the time-lag method may not be equal to the intrinsic diffusivity in the case of an incomplete liquid coverage on the upstream surface of the membrane. It is important to stress that the intrinsic diffusivity of the membrane remains constant and is independent of the fraction of the liquid coverage. For complete liquid coverage, the measured diffusivity should be equal to the intrinsic diffusivity.

### 2.3. Different Liquid Patterns and Surface Area Coverage

In this paper, several cases are examined to determine which variables influenced the values of the transport properties calculated for partial coverage scenarios. The fraction of coverage of the feed surface area of the membrane was varied to examine and quantify previous work carried out by Rivin et al. [11] which suggested that incomplete coverage leads to errors in the determined transport properties. The simulation also examined if different patterns of permeate added to the membrane surface would impact these measurements. Several patterns with different drop sizes and geometries were simulated and labelled 1 to 4 as illustrated in Figure 3. The droplet patterns were adapted from the test patterns described in US Army TOP 08-2-501A [8] to a droplet pattern with radial symmetry. Other drop patterns were also investigated.

## 3. Results and Discussion

This section presents the simulation results for various case studies to examine the impact of various variables on the estimation of the membrane properties, namely the liquid coverage pattern, fraction surface coverage, membrane intrinsic properties, and membrane thickness. The results show that the membrane properties are estimated accurately, as expected, in the case of a flooded cell and these results will serve as a basis of comparison for the other case studies.

### 3.1. Validation of the Numerical Simulations

Before presenting the results of various numerical simulations, it is important to examine the accuracy of the numerical model with a known benchmark solution. Figure 4 shows the comparison between the analytical and numerical solutions for the case where the feed side of the membrane is completely flooded at time *t* = 0. It is the only benchmark solution that is available for this problem, which is reduced in this case to a one-dimensional solution. Equation (13) gives the analytical solution of the concentration profile *c*(*z*,*t*) throughout the membrane as a function of time (*t*) and relative membrane thickness (*z*/*L*) for this case study. This analytical solution was obtained by the method of separation of variables [16]. The results of Figure 4 clearly show that the numerical solution for the concentration profile as a function of time corresponds exactly to the analytical concentration profile. An overall mass balance evaluated the accuracy of the simulations in the case of the partial liquid coverage of the membrane. In all cases, the overall mass balance was shown to be very accurate.
(13)$$c(z,t)=CS\left(1-\frac{z}{L}\right)-\left(\frac{2CS}{\pi }\right)\sum_{n=1}^{\infty }\left[\frac{1}{n}\sin\left(\frac{n\pi z}{L}\right)e^{-\left(\frac{Dn^{2}\pi^{2}t}{L^{2}}\right)}\right]$$

### 3.2. The Impact of the Membrane Diffusivity

To examine the impact of the membrane diffusivity (*D*) on the estimated effective diffusivity (*D*_eff_) for different membrane thicknesses, a series of numerical experiments were conducted over a wide range of membrane diffusivity values and for a constant liquid fraction coverage. The estimated diffusivity was obtained with the time-lag method. The results of these numerical experiments are presented in Figure 5 and show that, for a constant membrane thickness, the estimated effective diffusivity is directly proportional to the actual membrane diffusivity as the ratio *D*_eff_/*D* remains constant over the whole range of diffusivity. On the other hand, the thickness of the membrane has a major impact on the diffusivity ratio. For thin membranes, the diffusivity ratio is close to unity and the estimated diffusivity is relatively well estimated, where it approaches the value that is obtained with a flooded cell. It is important to state that the actual diffusivity (*D*) of the membrane is independent of the coverage pattern, and it is the value that we are attempting to recover. However, because of the fractional liquid coverage, the three-dimensional species permeation leads to an estimated value, termed effective diffusivity (*D*_eff_), that is different from the intrinsic membrane diffusivity.

### 3.3. The Impact of Membrane Thickness

Figure 6 presents the results of a series of simulations to examine the impact of the membrane thickness on the estimation of the membrane diffusivity for a single drop of liquid centrally located on the membrane and covering 18% of the membrane surface. The results show that, as the thickness of the membrane increases, the difference between the effective diffusivity and the actual diffusivity increases over a wide range of thicknesses. The diffusivity ratio (*D*_eff_/*D*) reaches a minimum at a thickness of approximately 6 mm with a relative diffusivity of approximately equal to 0.55. At higher membrane thicknesses, the effective diffusivity starts to increase. It is hypothesized that the diffusivity ratio will asymptotically increase back to unity for a much larger, albeit unrealistic, membrane thickness. Similar trends were observed for all the partial coverage patterns tested and the local minimum was in the same order of magnitude. This minimum occurs at smaller thicknesses for smaller coverage areas. These numerical simulations demonstrate that a significant estimation error of the intrinsic diffusivity should be expected for some common protective equipment [17,18,19,20], given the typical thickness of these protective garments. These errors typically lead to an underestimation of the actual diffusivity for a given system.

To explore the reason for the observed minimum in the diffusivity ratio and its subsequent increase as the thickness increases, the radial concentration profiles under steady state were obtained at five different relative depths (*z*/*L*) within the membrane and are presented in Figure 7. These results were obtained for a centered droplet covering 30% of the membrane surface area. Figure 7a presents the concentration profiles for a very thin membrane, *L* = 100 µm, showing very negligible radial diffusion such that the inlet and exit permeation areas are nearly identical with a flat concentration profile. The membrane diffusivity is very well estimated, with a ratio of *D*_eff_/*D* of 0.984. In Figure 7b, a larger radial diffusion was observed for a membrane having a thickness ten times larger (1.0 mm), which delayed the attainment of the steady state and the time lag. As a result, the estimated diffusion coefficient is smaller than the actual diffusion coefficient with a diffusivity ratio *D*_eff_/*D* of 0.809. When the membrane thickness is yet ten times larger (10 mm) and equal to the radius of the membrane, the migrating species have a significantly larger radial diffusion as shown in Figure 7c. The thickness is sufficiently large for the migrating species to diffuse to the impermeable diffusion cell boundary. As a result, a nearly flat concentration profile is observed close to the permeate side of the membrane. The estimated diffusion coefficient is smaller than the actual diffusion coefficient with a diffusivity ratio *D*_eff_/*D* of 0.643. This data point for this relatively thick membrane is in the increasing portion of the *D*_eff_/*D* versus *L* plot and further increasing the membrane thickness, although unrealistic, would lead to a relative diffusivity even closer to the actual diffusivity as a larger relative portion of the membrane reaches a flat concentration profile.

### 3.4. The Impact of Permeate Geometry

Figure 8 presents the results of the series of simulations performed to examine the variation in the relative diffusivity as a function of the thickness for two different drop patterns covering 12% of the surface membrane area. The droplet diameter, the diffusivity, and the number of droplets were held constant while, as shown in the two insets of Figure 8, the location of the outer layer of droplets was changed. The results show that, for thin membranes, the relative diffusivity ratio is approaching unity. However, as the thickness of the membrane increases, the relative diffusivity decreases rapidly to reach a minimum at a thickness between 1 and 2 mm. For both drop patterns, the minimum value of the relative diffusivity is nearly the same with a value of approximately 0.51 and 0.49 for the droplet pattern with the outer ring, shown in red, closer to the center and for the droplet pattern with the outer ring, shown in blue, further from the center, respectively. Beyond the minimum value, the relative diffusivity increases more smoothly with an increase in thickness. Even though the relative diffusivity is affected by the distribution of droplets on the surface of the membrane, the main factors affecting the diffusivity estimated by the time-lag method remain the thickness of the membrane and the liquid fraction coverage.

### 3.5. The Impact of Membrane Surface Area Covered by the Permeate

Figure 9 presents the results of a series of simulations performed to assess the variation in the relative diffusivity for two different liquid patterns, namely patterns 2 and 3, as a function of the surface coverage fraction. The results show that, as the fraction of the membrane surface covered by the liquid permeate increases, the relative diffusivity ratio approaches unity. Indeed, the intrinsic diffusivity of the membrane is obtained when the membrane surface is flooded by the liquid permeate. Results clearly show the effect of the liquid fraction coverage on the relative diffusivity as determined by the time-lag method. Both liquid patterns compared in this series of simulations are in close agreement with each other, suggesting that the effects of coverage are much less impacted by the geometry of the permeate on the membrane surface than the effects of thickness and the coverage fraction. Practically, in this case, as the fraction of the membrane covered by permeate increases, the patterns tested become increasingly similar.

Figure 10 shows a comparison of the relative diffusivity varying with the fractional coverage of the membrane surface for pattern 3 and four different membrane thicknesses. The results of Figure 10 clearly show the effects of both the coverage fraction and the membrane thickness for a constant coverage pattern. The value of the relative diffusivity *D*_eff_/*D* asymptotically tends to unity in all cases when the fraction coverage of the membrane surface area approaches unity, that is a flooded cell (pattern 1). These results corroborate the work of Rivin et al. [11] who state that the “flooded cell” scenario is the ideal method to estimate the intrinsic diffusivity of the membrane, as it corresponds to the case of one-dimensional diffusion.

### 3.6. Receding Liquid Coverage

Previous results clearly show that with flooded cell experiments, the membrane diffusivity is estimated accurately, whereas, for partial liquid surface coverage, its estimation needs special attention. In addition, with partial coverage, the liquid coverage area may recede as a function of time because of permeant sorption and evaporation, thereby affecting the dynamic permeation rate. Indeed, the rate of recession of the liquid typically depends on the thickness of the liquid cover, the liquid surface tension, the rate of evaporation, and its diffusivity and solubility in the membrane. Figure 11 presents the comparison of the dynamic permeation results for an initial central circular liquid pattern for a 2 mm thick membrane. A series of simulations were performed with different exponential decays of the liquid coverage radius as expressed by Equation (13), with the radius R of the liquid cover decreasing from the initial liquid radius R_0_ as a function of a receding factor α, time t, and time lag *θ*. The results shown in Figure 11 were obtained with an initial radius of approximately 0.007 m and a receding factor of 0.05 chosen to examine a slowly receding liquid.
(14)$$R=R_{0}e^{-\alpha \;t/\theta }$$

Figure 11a presents the inlet and outlet average fluxes of the migrating species as a function of time through the membrane. For a constant liquid cover (*α* = 0), the two average fluxes progressively become equal as they approach the steady state. When the liquid cover recedes as a function of time, as expected, the average fluxes become smaller than fluxes under constant coverage. For a longer permeation time, the liquid will completely disappear, and the two fluxes will eventually approach zero. For higher values of the receding factor, simulating higher diffusivities or lower vapor pressures, the average fluxes will decrease even faster. The corresponding amount of the migrating species that has permeated through the membrane is plotted in Figure 11b for the two values of the receding factor. For constant coverage, the rate of accumulation becomes relatively constant, and the time lag can be determined, while for receding coverage, the rate initially increases before progressively decreasing. For the latter case, it is more difficult to determine the time lag as the slope of the accumulation curve is never constant and does not entirely satisfy the underlying assumptions of the time-lag method. Figure 11c presents the estimation of the effective diffusivity determined from the instantaneous time lag as a function of time that was calculated from the slope of the plots of the accumulated amount that had permeated through the membrane (Figure 11b). In the case of constant liquid coverage, the instantaneous time lag progressively assumes a nearly constant value. A small decrease is still observed in Figure 11c because of the slow lateral diffusion that is required to reach a perfect steady state. For a receding liquid coverage, the instantaneous effective diffusivity initially decreases, then passes through a minimum before increasing again as a function of time. It is interesting to note that the minimum value of the effective diffusivity is very close to the actual value of the membrane diffusivity. In practice, a permeation experiment will probably last only for the first part of the curve when the concentration curve of the accumulated permeate would attain an almost linear trend, corresponding to an experimental time of between 1 and 2 × 10^6^ s (Figure 11b) for this relatively thick membrane. Finally, Figure 11d presents the variation in the radius of the central liquid coverage as a function of time for both cases considered in this section.

In practice, in the case of a receding liquid coverage to use the synergy between the experimental and numerical determinations of the effective diffusivity, it would be necessary to take images of the feed surface pattern as a function of time. It will then be possible to estimate the intrinsic diffusivity of the migrating species within the tested membrane.

### 3.7. Predictive Model for the Estimation of the Diffusivity Ratio

The results of the previous sections have clearly shown that a partial liquid coverage on the surface of the membrane may lead to an incorrect estimation of the diffusion coefficient of the liquid being tested. Ideally, a complete surface coverage should be used whenever possible. However, in the case of a very toxic liquid, it is advisable to use a drop pattern distribution to minimize the quantity of liquid used. The results obtained in this investigation provide the necessary data to develop a model that could predict the diffusion coefficient ratio *D*_eff_/*D*. The correction factor depends on numerous factors. Nevertheless, it is desired to find the simplest model that could provide an acceptable estimation of *D*_eff_/*D*. Having determined the effective diffusion coefficient *D*_eff_ experimentally, the actual diffusion coefficient *D* can therefore be estimated.

In this investigation, an artificial neural network was used to estimate the diffusion coefficient ratio as a function of the two most important contributing factors: the thickness of the membrane and the coverage fraction. Artificial neural networks are now commonly used successfully for a myriad of engineering applications. The high degree of plasticity in its structure is the main reason for its ability to efficiently represent the underlying causal relationship between input and output data. In this investigation, a three-layer feedforward neural network (FFNN) was used to predict *D_eff_/D* as a function of some input process variables. The FFNN is comprised of an input layer, a hidden layer, and an output layer, as shown in Figure 12. The input layer contains three neurons corresponding to the two independent variables plus the bias neuron. The two independent variables of Figure 12 are the scaled values of the logarithmic value of the membrane thickness and the liquid coverage fraction. The input information is simply transferred to the neurons of the hidden layer and the output layer. Appendix A provides the detailed development of the FFNN used to predict *D_eff_/D*. The results of 797 numerical simulations were used to determine the weights of the FFNN by minimizing the sum of squares of the differences between the diffusivity ratio *D_eff_/D* obtained numerically and the one predicted by the neural network. The 797 data points are comprised of 226 records for pattern 2, 254 for pattern 3, and 317 for pattern 4. Figure 13 presents the parity plot of the prediction of the diffusion coefficient ratio *D_eff_/D* for three different liquid coverage patterns. The coefficient of regression R^2^ is provided in Figure 13 for each liquid coverage pattern. The fit is not perfect since only the two main factors were considered as inputs to the model. The average relative error is 2.44% ± 2.36 and the maximum relative error is 15.4%. It is believed that the prediction error of this model is acceptable and allows us to estimate with good accuracy the diffusion coefficient ratio *D_eff_/D*.

## 4. Conclusions

This paper illustrates that the distribution of permeate on a membrane surface, which includes both the coverage of the surface area, and the pattern of species migration applied to the membrane, significantly influences the transport properties derived from permeation experiments. The effective diffusivity of the system varies with the fraction of the surface area of the membrane covered by the migrating species, the thickness of the membrane, and the geometry of the permeate applied to the membrane surface. For a flooded permeation cell, the intrinsic diffusivity is perfectly estimated. For membranes with partial liquid coverage, the effective diffusivity, as measured by the time-lag method, is lower than the intrinsic diffusivity of the membrane due to the multi-directional nature of the diffusion process. The experimental test methods used to evaluate PPE materials for protection against chemical warfare agents use partial liquid coverage. As a result, it may not be appropriate to use directly the transport properties obtained from these experimental methods to compare or generalize the protective performance of materials tested under different parameters such as material thickness, liquid coverage, and droplet pattern.

In instances where conducting experiments with flooded cells is either not feasible or impractical, applying a correction factor to the experimentally obtained transport properties can yield a more precise estimate of the intrinsic diffusivity. Subsequent experiments may be simulated using these intrinsic transport properties to investigate the impact of alterations in geometry and membrane thickness. Future research will involve conducting experiments to both assess and enhance the precision of our model, as well as to compare it against empirical data. Moreover, it would be interesting to investigate experimentally and numerically mixed-matrix or hybrid membranes to determine the estimation errors in their transport properties when characterized with the time-lag method. These newer membranes incorporate a reactive filler capable of neutralizing toxic chemicals as a means of detoxification [21,22,23].

## Figures and Tables

**Figure 1 membranes-14-00183-f001:**
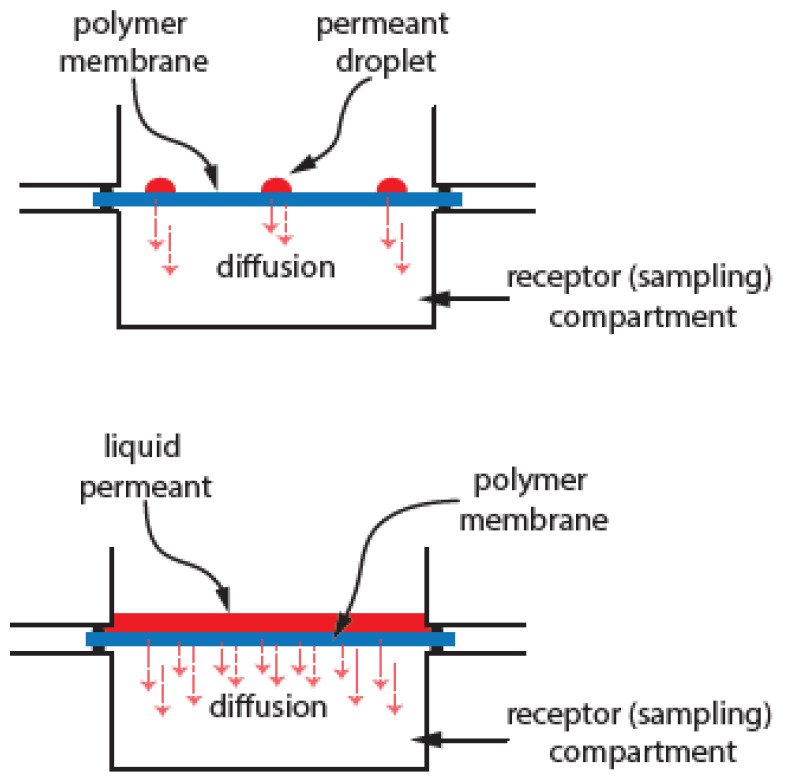
Simplified diffusion cell diagram depicting chemical permeant placed as droplets (**top**) or as flooded cell (**bottom**).

**Figure 2 membranes-14-00183-f002:**
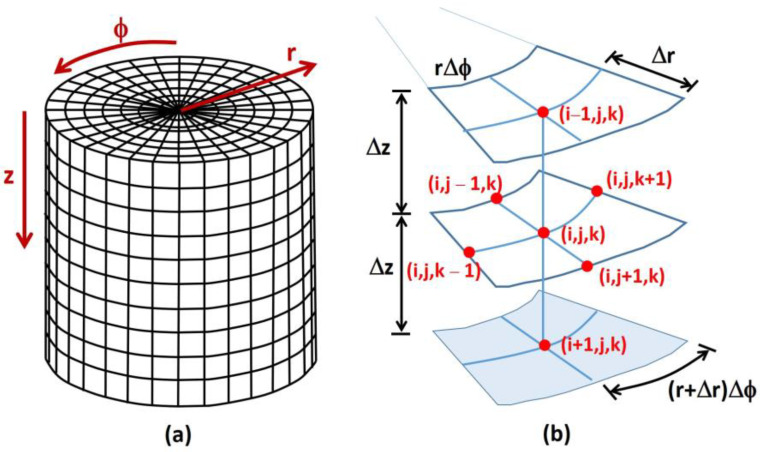
Discretized three-dimensional cylindrical membrane: (**a**) schematic diagram of a discretized membrane, and (**b**) central mesh point with its six neighboring mesh points used in Equation (2).

**Figure 3 membranes-14-00183-f003:**
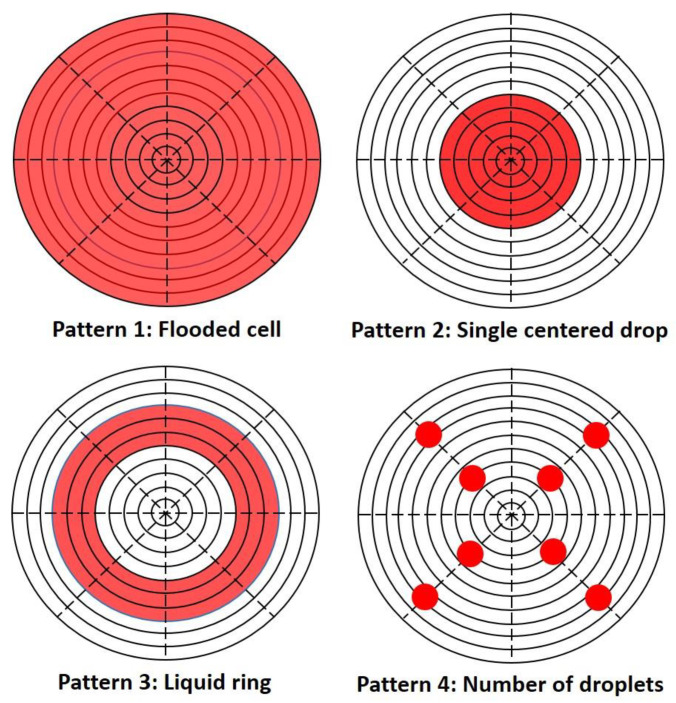
Examples of liquid patterns of the migrating species (in red) used at the membrane feed surface as boundary conditions for numerical experiments.

**Figure 4 membranes-14-00183-f004:**
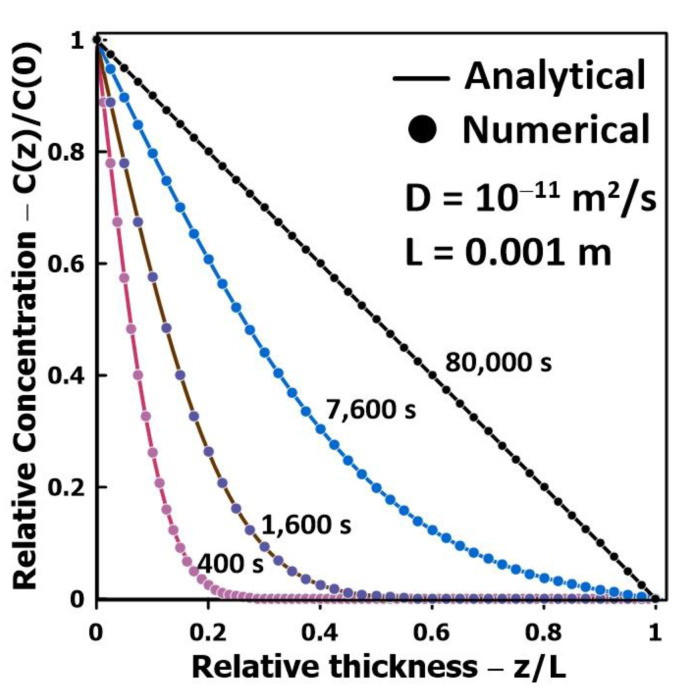
Comparison of analytical and numerical solutions in the case of one-dimensional diffusion occurring when flooded cell experiments are performed.

**Figure 5 membranes-14-00183-f005:**
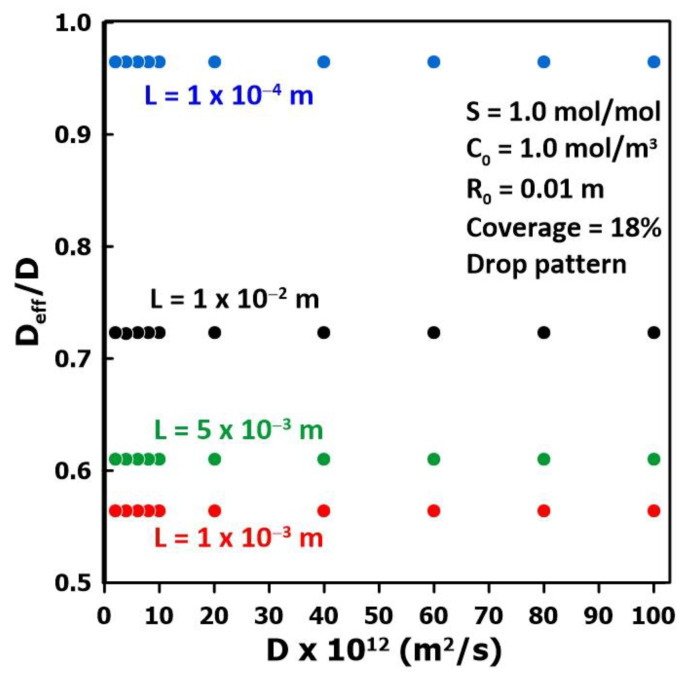
The ratio of the effective diffusivity and the actual diffusivity (*D*_eff_/*D*) as a function of the membrane diffusivity for four different membrane thicknesses with constant fraction coverage (pattern 4).

**Figure 6 membranes-14-00183-f006:**
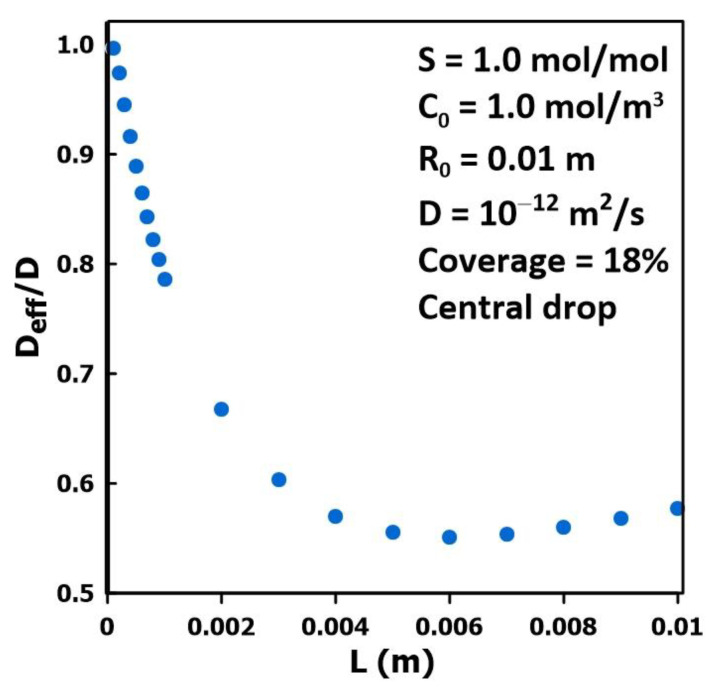
The diffusivity ratio (*D*_eff_/*D*) as a function of the membrane thickness for a central liquid drop (pattern 2) covering 18% of the membrane surface.

**Figure 7 membranes-14-00183-f007:**
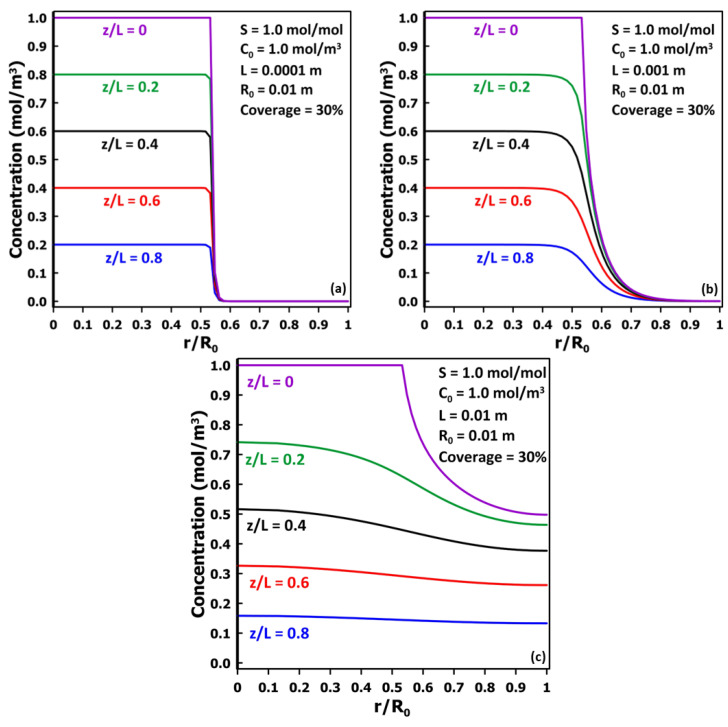
Concentration vs. *r*/*R*_0_ at different values of *z*/*L* for a single-centered droplet for three membrane thicknesses: (**a**) 100 μm, (**b**) 1.0 mm, and (**c**) 10 mm.

**Figure 8 membranes-14-00183-f008:**
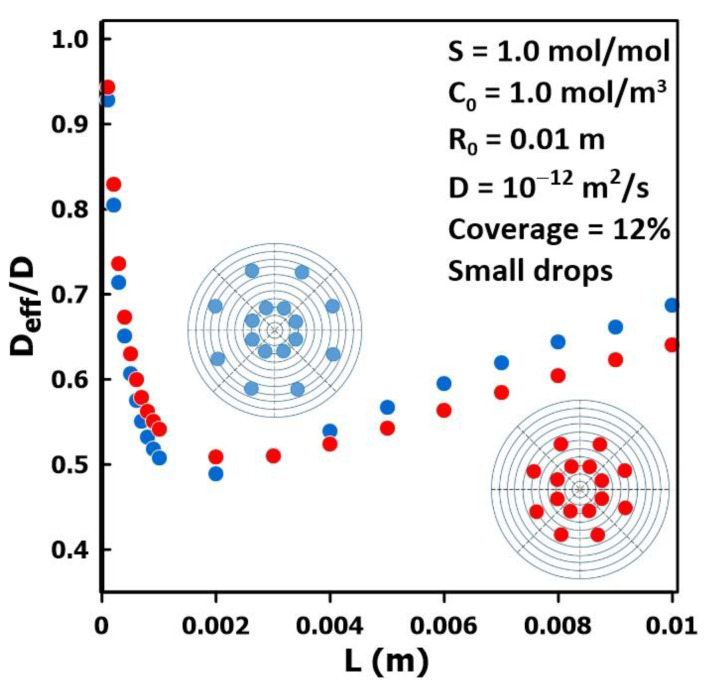
Diffusivity ratio *D*_eff_/*D* as a function of the membrane thickness for two different drop patterns for a liquid surface coverage of 12%.

**Figure 9 membranes-14-00183-f009:**
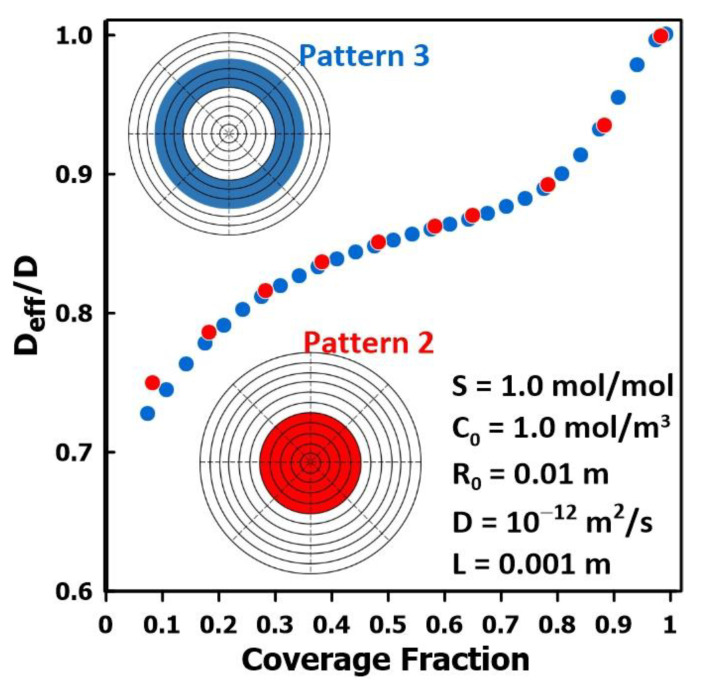
The ratio of the effective diffusivity over the actual diffusivity as a function of the liquid fraction coverage for patterns 2 and 3.

**Figure 10 membranes-14-00183-f010:**
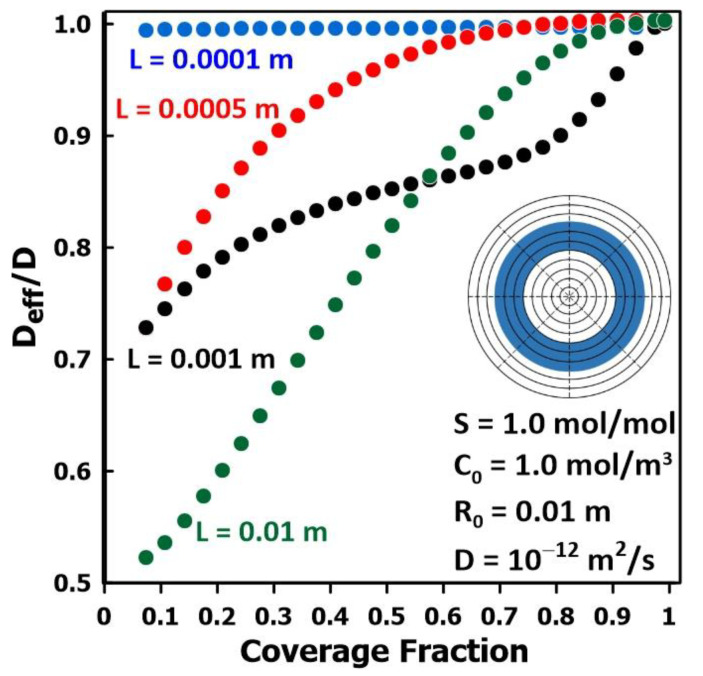
The ratio of the effective diffusivity over the actual diffusivity as a function of the coverage fraction for pattern 3, for four different membrane thicknesses.

**Figure 11 membranes-14-00183-f011:**
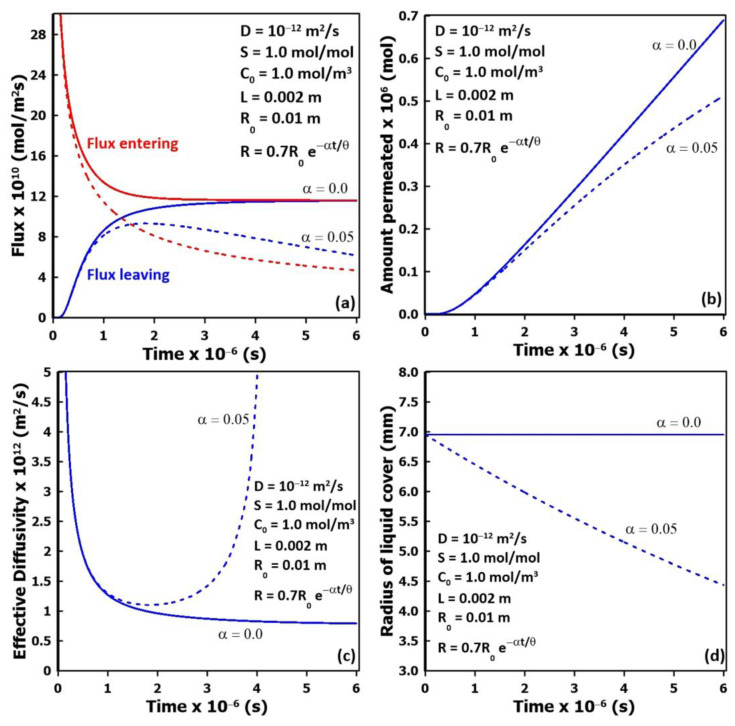
Comparison of the dynamic permeation results with (α = 0.05) and without (α = 0.0) receding liquid pattern: (**a**) upstream and downstream molar fluxes, (**b**) total amount permeated, (**c**) effective diffusivity estimated by the time-lag method, and (**d**) variation in the radius of the liquid coverage for the two receding factors.

**Figure 12 membranes-14-00183-f012:**
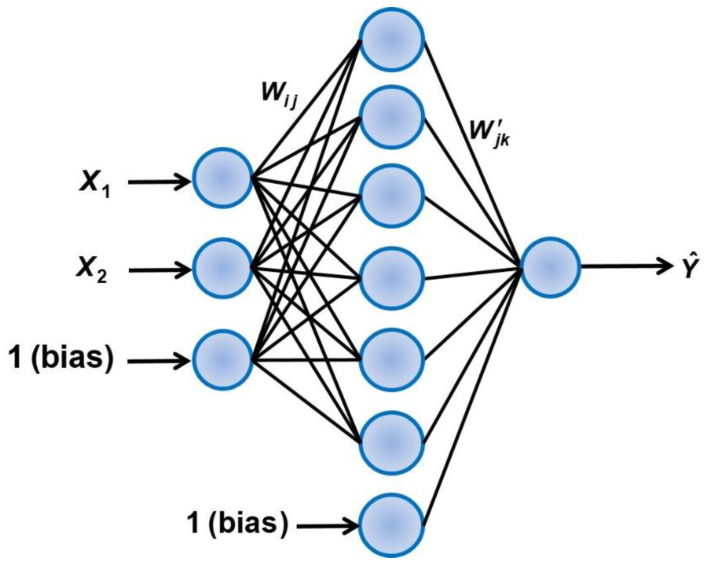
Feedforward neural network used for the estimation of the diffusion coefficient ratio *D*_eff_/*D*.

**Figure 13 membranes-14-00183-f013:**
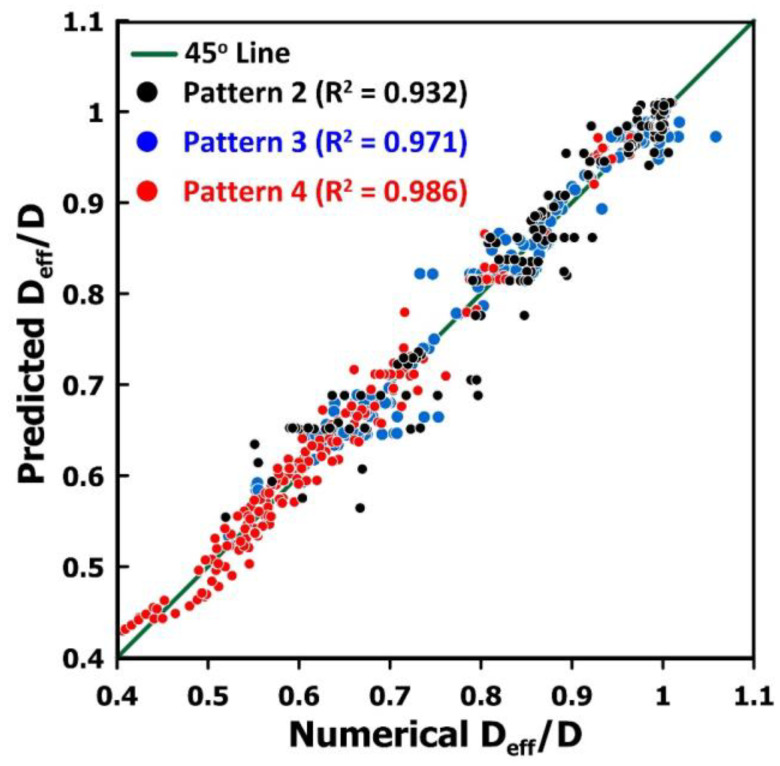
Parity plot of the prediction of the diffusion coefficient ratio *D*_eff_/*D* for three different liquid coverage patterns and the FFNN of Figure 12.

## Data Availability

Data are available upon request.

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
