# Peer review of "Simulating the Permeation of Toxic Chemicals through Barrier Materials"

_membranes, 2024, doi:10.3390/membranes14090183_

Round 1

Reviewer 1 Report (New Reviewer)

Comments and Suggestions for Authors

The author of this article does a very good job of explaining the formulas clearly and logically. They also describe the experimental settings and numerical simulations. They also discuss boundary conditions and how to deal with the initial conditions of the simulation.

However, the minor issues below still need to be corrected. Therefore, I recommend giving the article a Minor Revision based on the Authors' reply before publishing it.

a.      LINE 108. The explanations of Formula 1 and Formula 2 here are well written, but it is recommended to place them below each formula and explain them separately to make it easier to read.

b.      LINE 179. The picture here should be Figure 3, not Figure 1. This leads to errors in the encoding of the image below. Although it does not affect the presentation of the results, it makes the article very unsound to read.

c.      LINE 418. At the conclusion stage, the author quoted other articles' conclusions. This is inappropriate. Please rewrite.

d.      A large part of the author's experimental results and discussion echo the study by Rivin et al. in 2005. The authors are asked to explain the reasons. The results explained in this way will largely determine the novelty of this article.

e.      Although the author uses the FFNN prediction method, we usually train FFNNs through back-propagation. Still, the author uses a forward training method, which differs from today's method. Please explain.

Author Response

The authors would like to thank the reviewers for their generous comments and suggestions that have helped improve the manuscript's quality. We have tried our best to respond to all comments and questions.

We have answered the comments in the same order provided by the reviewers. The reviewers' comments are given in black and the answers are in blue.

Reviewer 1:

The author of this article does a very good job of explaining the formulas clearly and logically. They also describe the experimental settings and numerical simulations. They also discuss boundary conditions and how to deal with the initial conditions of the simulation.

However, the minor issues below still need to be corrected. Therefore, I recommend giving the article a Minor Revision based on the Authors' reply before publishing it.

  1. LINE 108. The explanations of Formula 1 and Formula 2 here are well written, but it is recommended to place them below each formula and explain them separately to make it easier to read.

Eqs. (1) and (2) were moved upward in the text, i.e. immediately after they were introduced, as suggested by the reviewers.

LINE 179. The picture here should be Figure 3, not Figure 1. This leads to errors in the encoding of the image below. Although it does not affect the presentation of the results, it makes the article very unsound to read.

In the submitted manuscript that I downloaded from the system, Figure 3 appears in the figure caption. The error may be due to the auto-referencing system that is used. To avoid this confusion, I have removed the auto-referencing system for all figures by rewriting the figure number. I hope the last version will appear correctly.

LINE 418. At the conclusion stage, the author quoted other articles' conclusions. This is inappropriate. Please rewrite.

I read carefully our conclusion to detect where the conclusions of other papers were written. I did not detect concluding remarks from other papers. I suspect that the reviewer is referring to the last paragraph where future works are discussed. To avoid confusion the last part of this paragraph was rewritten as follows: “Future research will involve conducting experiments to both assess and enhance the precision of our model, as well as to compare it against empirical data. Moreover, it would be interesting to investigate experimentally and numerically mixed-matrix or hybrid membranes to determine the estimation errors in their transport properties when characterized with the time-lag method. These newer membranes incorporate a reactive filler capable of neutralizing toxic chemicals as a means of detoxification [21-23].” References are used to give credit to the authors who have developed these novel membranes that we would like to examine in the context of the current paper. We are not using their conclusions.

A large part of the author's experimental results and discussion echo the study by Rivin et al. in 2005. The authors are asked to explain the reasons. The results explained in this way will largely determine the novelty of this article.

The main reason for referencing Rivin et al. (2005) is mostly due to the domain of application. Rivin works for the U.S. Army Research Development and Engineering Command. He is interested in the determination of the transport properties of materials subjected to different chemical agents. However, the two studies are completely different. We refer to Rivin because he was the one, and the only one to our knowledge, that attracted attention to the potential error in the estimation of transport properties when droplet patterns are used, however, without quantifying the magnitude of the error. Our paper quantifies the estimation errors. In his paper, Rivin proposed a new permeation cell where a flooded membrane can be used safely. This means obviously using a larger quantity of liquid when performing experiments.

Although the author uses the FFNN prediction method, we usually train FFNNs through back-propagation. Still, the author uses a forward training method, which differs from today's method. Please explain.

FFNN models are trained by repeatedly exposing the model to examples of input and output and adjusting the weights to minimize the error of the model's output compared to the expected output. The optimization method used to adjust the weights of the FFNN is the quasi-Newton method. Some authors use many other optimization algorithms such as Marquart, backpropagation, ADAM, etc. I found for a relatively small databank and small size of the FFNN structure, a more powerful method leads more rapidly to an accurate model. I have clearly shown in a paper the drastic difference between the backpropagation and the quasi-Newton method (J. Thibault, Feedforward neural networks for the identification of dynamic processes, Chem. Eng. Comm. 1991, Vol. 105, pp. 109-128). Even though the initial backpropagation has been improved with the ADAM algorithm, it remains a very slow method. It is of course used for large neural networks and large databanks, where quasi-Newton would then take more time to converge. In this investigation, I can guarantee that quasi-Newton outperforms backpropagation.

Reviewer 2 Report (New Reviewer)

Comments and Suggestions for Authors

This manuscript left me in mixed feelings. I understand that a substantial amount of work has been done in writing the code and analyzing the results, but solving diffusion equation in cylindrical symmetry is no more news. For example, Crank's famous textbook 'The Mathematics of Diffusion' has an entire chapter of that. Numerical solutions have been treated in, e.g. Dieter Britz's book 'Digital simulation electrochemistry' - mass transport is always coupled to electrochemical work. Hence, the novelty remaining is the partial coverage of the diffusing substance. For that I would use Comsol Multiphysics but, naturally, writing own code gives a chance to control the simulations in a more versatile way.

I hail fundamental work without imminent applications but the authors' justification of their study with toxicology is not really coupled to the core study by any means; the results apply to any diffusion process. But if we wanted to emphasize occupational safety, the scope could be wider. Now the model concerns only non-porous membranes where, e.g. capillary forces do not play any role. In practice, diffusion is not the only mechanish with which toxic chemicals can penetrate, say, gloves.

What comes to calculations, it is common to make variables dimensionless in order to make simulations more general. Concentrations are scaled with the initial (bulk) concentration, spatial coordinates in this case with the membrane thickness, L, and the dimensionless time as T = Dt/L². Upon these conversions, D, L and C(0) disappear entirely from equations, and figures like Fig. 2 - that I present in my bachelor course - do not need to define explicitly the value of, say, the diffusion coefficient. But I believe that the calculations are correct.

The only unclear issue is the effective diffusivity. I think that the authors did not reply properly to the reviewer's question in the previous round. Hence, please define explicitly what it is. Figure 3 manifests this dilemma. Eq. (12) is a well-known expression for the lag-time in a 1-D case, derived from eq. (13). Now, are you stating that eq. (12) applies in all studied cases by just adjusting the value of Deff?

Author Response

The authors would like to thank the reviewers for their generous comments and suggestions that have helped improve the manuscript's quality. We have tried our best to respond to all comments and questions.

We have answered the comments in the same order provided by the reviewers. The reviewers' comments are given in black and the answers are in blue.

Reviewer 2:

This manuscript left me in mixed feelings. I understand that a substantial amount of work has been done in writing the code and analyzing the results, but solving diffusion equation in cylindrical symmetry is no more news. For example, Crank's famous textbook 'The Mathematics of Diffusion' has an entire chapter of that. Numerical solutions have been treated in, e.g. Dieter Britz's book 'Digital simulation electrochemistry' - mass transport is always coupled to electrochemical work. Hence, the novelty remaining is the partial coverage of the diffusing substance. For that I would use Comsol Multiphysics but, naturally, writing own code gives a chance to control the simulations in a more versatile way.

I agree that solving the three-dimensional Fick’s second equation of diffusion has no novelty. The paper does not claim any novelty in solving this equation. The numerical solution is simply a tool to determine the significant error occurring in the estimation of the transport properties when a drop pattern or partial liquid coverage is used in conjunction with the time-lag method. Using COMSOL will also not add a degree of novelty – it is used in our undergraduate classes to solve heat transfer problems. Whether a homemade finite difference algorithm or COMSOL is used, the problem still needs to be defined with the presentation of the equations to be solved, the initial condition, and all the boundary conditions as it was done in this paper. Possibly only Eq. (2) would not appear in the paper. The novelty is indeed the impact of the partial coverage of the diffusing substance on the estimation of the transport properties. The authors feel that it is a significant problem to address.

I hail fundamental work without imminent applications but the authors' justification of their study with toxicology is not really coupled to the core study by any means; the results apply to any diffusion process. But if we wanted to emphasize occupational safety, the scope could be wider. Now the model concerns only non-porous membranes where, e.g. capillary forces do not play any role. In practice, diffusion is not the only mechanish with which toxic chemicals can penetrate, say, gloves.

The study could indeed be much wider where other materials used for personal protection equipment (PPE) could be considered. However, we need to start somewhere and this study considered barrier materials, that are non-porous membranes. This study aimed to provide some guidance to research laboratories that use the time-lag method with drop patterns to determine the transport properties of materials used for PPE. Rivin et al. (2005) have mentioned that the transport properties may not be evaluated accurately, but without quantifying the magnitude of the errors. This was the objective of this investigation. If non-porous materials were considered, other permeation mechanisms would need to be considered such as Poiseuille flow, Knudsen diffusion, surface diffusion, capillary condensation, molecular sieving, solution-diffusion, etc. This particular problem could be considered in another paper. The current investigation is restricted to barrier materials where solution-diffusion mechanism applies.

What comes to calculations, it is common to make variables dimensionless in order to make simulations more general. Concentrations are scaled with the initial (bulk) concentration, spatial coordinates in this case with the membrane thickness, L, and the dimensionless time as T = Dt/L². Upon these conversions, D, L and C(0) disappear entirely from equations, and figures like Fig. 2 - that I present in my bachelor course - do not need to define explicitly the value of, say, the diffusion coefficient. But I believe that the calculations are correct.

It is true that dimensionless variables are commonly used with great advantages for these types of numerical methods. However, the use of dimensionless variables only applies to one-dimensional problems. For multidimensional problems, it is not possible to define dimensionless time as more directions are involved; as a result, the time and dimensions must remain in their native units. It is only possible to use a dimensionless concentration, but not dimensionless space and time.

The only unclear issue is the effective diffusivity. I think that the authors did not reply properly to the reviewer's question in the previous round. Hence, please define explicitly what it is. Figure 3 manifests this dilemma. Eq. (12) is a well-known expression for the lag-time in a 1-D case, derived from eq. (13). Now, are you stating that eq. (12) applies in all studied cases by just adjusting the value of Deff?

It is important to note that a typical dynamic test using a permeation cell with the time-lag method leads to an estimation of the membrane diffusivity, referred to in the manuscript as Deff. The latter is an apparent or effective diffusivity because it may not be equal to the intrinsic diffusivity. Indeed, when the liquid coverage on the upstream side of the membrane is only partial, the time-lag method may not lead to the actual value of the diffusivity. Because the actual or intrinsic diffusivity is constant and independent of the fraction of the liquid coverage, it is possible to infer from the measured (effective, apparent) diffusivity to estimate the intrinsic diffusivity. This is one the purposes of this investigation. To enhance the clarity of this potential confusion, the following text has been added in the paragraph following Eq. (12): “(Lines 166-172) Note that an apparent or effective diffusivity (Deff) has been used Eq. (12) instead of the intrinsic diffusivity (D) because the measured diffusivity via the time-lag method may not be equal to the intrinsic diffusivity in the case of an incomplete liquid coverage on the upstream surface of the membrane. It is important to stress that the intrinsic diffusivity of the membrane remains constant and is independent of the fraction of the liquid coverage. For complete liquid coverage, the measured diffusivity should be equal to the intrinsic diffusivity.”

Round 2

Reviewer 2 Report (New Reviewer)

Comments and Suggestions for Authors

The authors have now disclosed how the effective diffusivity was determined; it is via eq. (12) as I expected. Please add a reference to eq. (13).

Their reply that dimensionless forms can be used only in 1-D simulations is simply incorrect, it is done all the time.

Author Response

This manuscript is a resubmission of an earlier submission. The following is a list of the peer review reports and author responses from that submission.

Round 1

Reviewer 1 Report

Comments and Suggestions for Authors

In this study, the authors present a numerically solution of the three-dimensional cylindrical Fick´s second law of diffusion for a liquid permeating through a non-porous rubbery membrane to determine the time the permeating species will emerge on the other side of the polymer membrane. The manuscript is very interesting, well-articulated, presenting well-designed figures and mathematically validated. Overall, this manuscript is recommended to be published in this journal after a minor revision.

  1. I recommend that the authors prepare a comprehensive figure to be placed at the beginning of this manuscript, which should encapsulate the entirety of the research process. This figure could be similar to a graphical abstract, detailing each step procedure. This will enhance the reader's comprehension of the manuscript. It will be particularly beneficial for readers who may not be familiar with mathematical models, demonstrating how they can apply these findings to their own work.

  1. In Section Materials and Methods, specifically sub-section 2.1 (Experimental Permeation System), I suggest that the authors include detailed information regarding the membrane and liquids (among others) utilized in the experiment. This should encompass specifics such as the material of the membrane, the manufacturer responsible for producing it, and any unique identification numbers associated with it. Please ensure that the entire manuscript is thoroughly reviewed to include all essential information necessary to accurately replicate these experiments.

  1. I recommend that the authors provide details about the software utilized to process the raw data for this mathematical model, as well as the statistical methods applied, and how many replicates the authors did per condition.

  1. To finalize, I suggest a comprehensive revision of the entire manuscript to address typographical errors and language issues. 

Reviewer 2 Report

Comments and Suggestions for Authors

Chapters 3.2 and 3.3 should be reconsidered by the authors.  The diffusion coefficient depends on temperature, pressure or concentration. In this paper, the authors try to relate the value of the diffusion coefficient to the thickness of the membrane, which is an incorrect assumption. The diffusion coefficient can be constant, but can also depend on time D(t) or concentration D(C). The authors should first try to consider how different amounts of substances represented by different liquid patterns at the membrane feed surface affect/change the boundary conditions considered in the simulation. This issue should be further described and discussed.

Additional minor comments:

1. Authors should avoid citations in the abstract. 

2. Page 2 (lines 91-92) and page 3 (lines 102, 110) There is :  Error! Reference source not found. 

3 Chapter 2.2 - beginning - text error.